# HIERARCHICAL LONG-TAILED CLASSIFICATION WITH VISUAL LANGUAGE MODELS

## ABSTRACT

Vision Language Models (VLMs) have shown promising capabilities in handling open vocabulary tasks but struggle with imbalanced data tuning, particularly when dealing with highly skewed label distributions. To address the challenges, we propose a hierarchical long-tailed classification framework, named HLC, which prioritizes candidate categories before conducting fine-grained classification using detailed textual descriptions. Specifically, we fine-tune a linear classifier based on the CLIP encoder, incorporating visual prompt tokens and leveraging shared feature space mixup for multimodal feature interactions. Based on candidates given by the coarse classifier, we query large language models to generate corresponding fine-grained descriptions to refine the final predictions. Importantly, we introduce a reweighting mechanism to filter out invalid descriptions generated by language models. Extensive evaluations demonstrate that our approach achieves state-of-the-art performance by fine-tuning only a few parameters on the PlacesLT, ImageNet-LT, and iNaturalist 2018 datasets.

## 1 INTRODUCTION

Real-world visual data typically exhibits an instance-imbalanced long-tailed distribution. Models trained on such skewed datasets over-focus on the majority (*head*) classes while neglecting the minority (*tail*) ones, resulting in the bias for the head and poor generalization on the tail (Liu et al., 2019; Cui et al., 2019; Xu et al., 2021; Cao et al., 2019). Researchers struggle to alleviate the Long-Tailed Recognition (LTR) problem by leveraging visual datasets alone to train classifiers with elaborately designed strategies, briefly taxonomized into four categories: 1) rebalancing by reweighting (Cui et al., 2019; Xu et al., 2023b; Ma et al., 2023) or resampling (Cao et al., 2019; Kang et al., 2020), 2) enhancing the tail with head categories (Chou et al., 2020; Park et al., 2022) 3) decoupling feature learning and downstream tasks with two-stage framework (Kang et al., 2020; Zhou et al., 2023), and 4) integrating multi-experts to focus on different aspects (Li et al., 2022a; Jin et al., 2023; Xu et al., 2023a). In this paper, we observe that *while previous methods can achieve satisfactory top-5 classification accuracy, the true challenge lies in fine-grained predictions from the candidates (Figure 1a)*. However, naively enumerating possible fine-grained classifiers and training each candidate yield exponential computational overheads.

The recent success of Open-Vocabulary Classification (OVC) can fine-tune a few parameters of the pre-trained VLMs, such that the classifier can be ready for arbitrary category numbers with satisfactory Few-Shot Learning (FSL) capabilities. One may intuitively resort to the versatile large-scale vision language models (VLMs) (Radford et al., 2021; Alayrac et al., 2022; Jia et al., 2021) as auxiliary to support image classification. Is it possible to embrace OVC to compensate the inherently imbalanced datasets without extreme computational overheads? Unfortunately, while OVC excels at handling fine-grained recognition with label prompts (Yao et al., 2021), it is incapable of catching up with the state-of-the-art approaches (Table 1) in the challenging LTR, because the OVC fails to 1) take advantage of existing annotation information; 2) overcome the long-tail bias problem; and 3) fully encapsulate the visual information with existing language description.

In this paper, we make an affirmative answer that VLMs are strong enough to promote state-of-the-art approaches in LTR, by proposing tailored solutions to address the above issues. We follow previous methods (Tian et al., 2022; Zhou et al., 2022b; Pratt et al., 2022) to collect class-wise corpora from the Web, such that the LTR datasets (Liu et al., 2019) are equipped with reasonable language

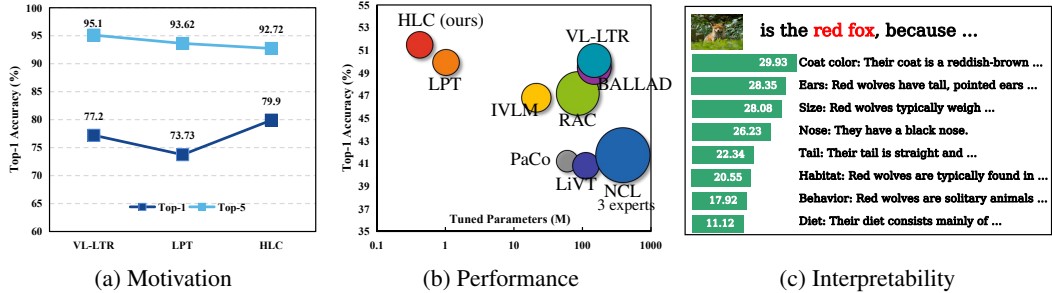

(a) Motivation            (b) Performance            (c) Interpretability

Figure 1: a) Methods performance on ImageNet-LT. b) Model performance *v.s.* Tuned parameters on PlacesLT. The radius is the model size. c) Our HLC conducts inference with interpretability.

information to conduct mix-modality tuning. We then prompt the Large Language Models (LLMs) to generate fine-grained descriptions for each class and propose the novel Hierarchical Long-tailed Classification (HLC) framework, which combines the traditional coarse classifiers and OVC in a hierarchical manner. After obtaining *top-k* candidate categories from the trained coarse classifier in the first stage, instead of training numerous candidate classifiers, we alternatively employ OVC for further fine-grained classification without parameter tuning (Figure 1b).

Our HLC is composed of several crucial components. *First*, we integrate the advantages of the abovementioned class-wise corpora to fine-tune additional visual prompt tokens (Jia et al., 2022) and the coarse classifier. In this way, the model trained with mixed-modal data absorbs the reasonable features by Vision-Language (V-L) shared space compared to naive image space. *Second*, given the empirical observation that the classifier performs better on text than on visual data, we propose Shared Feature space Mixup (SFM) to enhance the correspondence of multi-modality data and specialize HLC for LTR. Further, we adopt the post-hoc logit adjustment (Menon et al., 2021) to eliminate the preference for the head and improve the robustness of the tail. *Third*, considering the inevasible mismatched descriptions given by LLMs, we propose adaptive weights tuning for each description and jointly optimize weights with coarse classifier and visual prompt tokens, such that crucial descriptions are emphasized while negligible parts are weakened. Once trained, we calculate the expectations of all weighted image-descriptions similarity to integrate the final prediction for explainable reasoning (Figure 1c). We present extensive experiments to demonstrate the advances of the proposed method, with detailed ablation studies to manifest the effectiveness of our proposals.

In summary, our contributions are three-fold. *First*, we supplement class-wise corpora for the missing text information in large-scale LTR benchmarks and leverage the versatile LLMs to construct informative and detailed descriptions of each category. *Second*, we propose novel HLC for OVC to perform fine-grained recognition in LTR, with two tailored solutions, *shared feature space mixup* and *adaptive weights tuning* mechanism, to prevent head-prejudice and side-effect of class-irrelevant descriptions. *Finally*, with the organic integration of these crucial insights and techniques, we demonstrate the state-of-the-art performance of OVC with HLC on challenging imbalanced benchmarks *w.r.t.* Places-LT, ImageNet-LT, and iNaturalist 2018. Our class-wise corpora, descriptions from LLMs, and models will be publicly available for research purposes.

## 2   PRELIMINARIES

**Task Definition.** We focus on the LTR task, where the training data lies in long-tailed distribution *w.r.t.* the class labels. Given a visual dataset with $C$ classes, $\mathcal{D}_{\mathcal{V}} := \{(x_i^{\mathcal{Y}}, y_i)\}_{i=1}^{N}$, where $x_i^{\mathcal{Y}} \in \mathbb{R}^{H \times W \times 3}$ and $y_i \in \mathbb{R}^C$, we denote the instance number of the $i$-th class as $n_i$ and $n_1 \geq n_1 \geq ... \geq n_C$, where $n_1 \gg n_C$ typically in LTR.

**Vision Language Model.** Our proposals are based on pre-trained vision-language models, *e.g.*, CLIP, with the ViT-B as the visual encoder $\mathcal{E}^V$. The language encoder $\mathcal{E}^L$ maps the textual corpora into the visual language (V-L) shared feature space $v^{V-L} \in \mathbb{R}^d$. For the visual branch, given a query image $I \in \mathbb{R}^{H \times W \times 3}$, the visual encoder $\mathcal{E}^V$ splits the image $I$ and embedding it into $M$ patches tokens $E_0$. Combined with a CLS token $c_0$, we get the input visual token sequence $[E_0, c_0]$, which will be processed by $L_t$ transformer layers. The output CLS token $c_{L_t}$ of the final transformer layer will be projected to V-L shared latent embedding space.

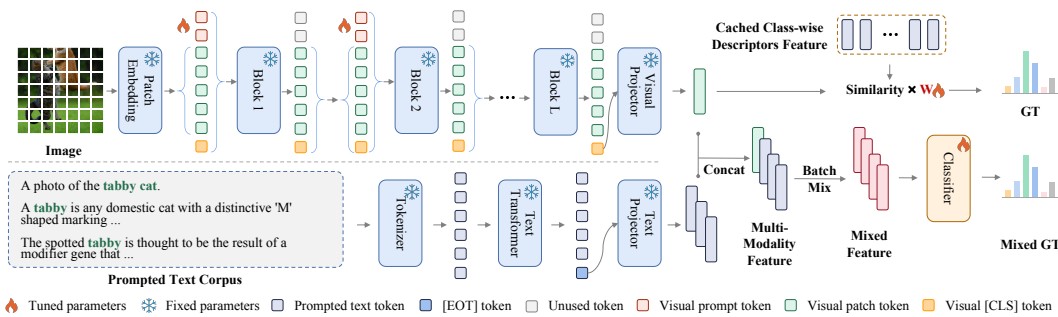

Figure 2: The finetuning pipeline. 1) We conduct mixed training with collected category-specific corpora. 2) We adopt visual prompt tuning to better fit the training data. 3) We propose V-L shared feature space *mixup* to increase the interaction between the two modalities' features. 4) We construct a fine-grained text description feature cache and optimize the weight of each description ($M = 10$) to ignore the impact of the training image-unrelated descriptions.

## 3 METHODOLOGY

### 3.1 CROSS-MODAL TRAINING.

Although the open-vocabulary classification is promising, the performance is still far from the SOTA on the LTR benchmarks, *e.g.*, zero-shot CLIP 37.9% *v.s.* VL-LTR 50.1% on the PlacesLT in Table 1. Hence, we attempt to amalgamate prior research with OVC to leverage the VLMs' multimodal capabilities. Initially, we aim to train a high-performance classifier that can yield reliable candidates. We consider a cosine classifier $W^{cls}$ and the training framework is shown in Figure 2. We utilize the shared feature space of CLIP (Radford et al., 2021) to assist the classifier training because the mixed-modal data can benefit downstream uni-modality tasks (Lin et al., 2023). Some tail images are challenging to collect, while the relevant textual descriptions can be obtained from the Internet easily. Therefore, we construct corpora to supplement the few-shot visual features and train the classifier with mixed modality data (See section 4.1 for details). Given mini-batch input images $x^V$ and texts $x^L$, the empirical risk minimization will be formulated as follows:

$$\hat{y} = \arg\max \frac{W^{cls} \cdot v^{V-L}}{||W^{cls}||_2 \cdot ||v^{V-L}||_2}, \quad (1)$$

where $v^{V-L} := \texttt{concat}(\mathcal{E}^V(x^V), \mathcal{E}^L(x^L))$ is the feature batch given by the CLIP encoders.

Experimentally, the classifier $W^{cls}$ performs much better on textual modality than on images, with slight improvement in the tail classes. Hence, we propose the Shared Feature space *Mixup* (SFM) to enhance the feature interaction between the two modalities (see Figure 2). Give the corresponding ground-truth labels $y^V, y^L$, the embedding-level *mixup* will be:

$$\begin{aligned} v^{V-L}_{\text{mixed}} &= \lambda \cdot v^{V-L} + (1 - \lambda) \cdot \phi(v^{V-L}) \\ y_{\text{mixed}} &= \lambda \cdot y + (1 - \lambda) \cdot \phi(y), \end{aligned} \quad (2)$$

where $y := \texttt{concat}(y^V, y^L)$, $\lambda$ is sampled from Beta distribution and the $\phi(\cdot)$ is the batch shuffle operation. Note that the mixing operation takes place in the shared space rather than at the input level in vanilla *mixup* (Zhang et al., 2017), thereby avoiding the contradiction arising from disparate modalities. While MixGen (Hao et al., 2023) attempts to directly concatenate text to achieve input-level *mixup*, our experimental results suggest that embedding-level *mixup* is more effective in training high-performance classifiers when the ground-truth labels are available.

Considering the similarity between the few-shot learning and tail learning of LTR, we employ deep Visual Prompt Tuning (VPT) (Jia et al., 2022) in the few-shot learning area to adapt the downstream data distribution. Concretely, we introduce a group of learnable visual prompts tokens $\tilde{p} := [p_0, p_1, \cdots p_{N-1}]$ to build the input sequence of the visual branch and more learnable tokens $\{\tilde{p}_i\}_{i=0}^{L_p}$ will be introduced in deeper transformer layers up to depth $L_p$. Hence, we derive the token sequence and the toke flow in the vision branch will be:

$$\begin{aligned} [\_, E_i, \mathsf{c}_i] &= \mathcal{L}_i([\tilde{p}_{i-1}, E_{i-1}, \mathsf{c}_{i-1}]) & i = 1, 2, \cdots, L_p. \\ [E_j, \mathsf{c}_j] &= \mathcal{L}_j([E_{j-1}, \mathsf{c}_{j-1}]) & j = L_p + 1, \cdots, L_t. \\ v^{V-L} &= \texttt{ImageProj}(\mathsf{c}_{L_t}) & v^{V-T} \in \mathbb{R}^d. \end{aligned} \quad (3)$$

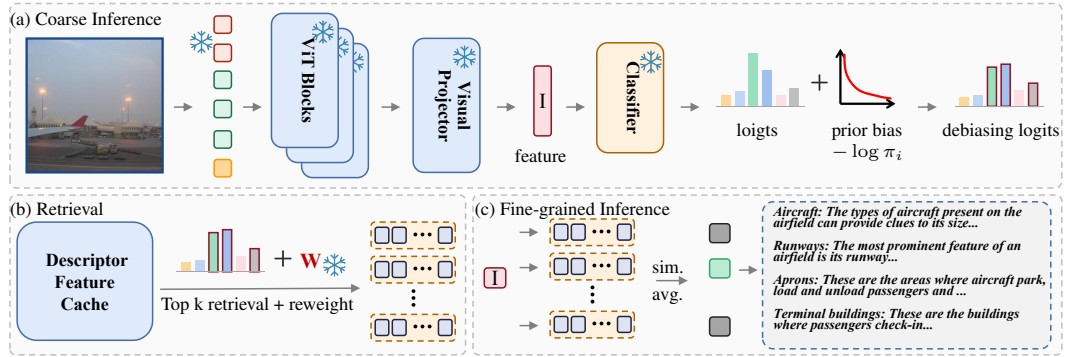

Figure 3: The inference pipeline. 1) We perform a coarse-grained inference to obtain features and debiased logits. Post-hoc adjustment is used for long-tail debiasing. 2) We retrieve the descriptions features and corresponding weights of the *top k* candidates based on the debiased logits. 3) We calculate the weighted average similarity between the image and $M$ text descriptions feature of candidate classes. The results have good interpretability with matching scores of different descriptions.

## 3.2 RECOGNITION WITH WEIGHTED DESCRIPTORS.

To perform fine-grained classification, we further construct descriptor sets that characterize $M$ detailed features of the category. Following VCD (Menon & Vondrick, 2023), we prompt large language models, *e.g.*, GPT-3.5-turbo, to generate fine-grained descriptions to verify class labels. To match the fine-tuning data, we let LLMs generate sentence-level descriptions instead of phrases in VCD. Let's take the query prompt of PlacesLT (Liu et al., 2019) as an illustrative example:

```
Q: List 10 useful features for distinguishing {CLS} in a photo.
```

In contrast to VCD, we observe that LLMs tend to provide descriptions that are unrelated to the images in the datasets (see examples in Figure 6). To mitigate their impact on the overall performance, we introduce a set of learnable weights $W^{llm}$ for each descriptor and update parameters during classifier training (see Figure 2). The effectiveness of our descriptions and the adaptive weights mechanism has been demonstrated to be superior to VCD by ablation experiments (Table 7). In contrast to previous methods that employ the text encoder in intricate loss optimization processes (Zhou et al., 2022a; khattak et al., 2023), our approach relies solely on a single forward pass of the text encoder for all descriptions and caches the corresponding features. This allows us to perform open-vocabulary classification without incurring any additional computational overhead.

## 3.3 PIPELINE.

Based on the aforementioned proposals, we present our hierarchical LTR framework. First, we train a base classifier with SFM and VPT. To mitigate bias incurred by long-tail data, we further adopt the post-hoc logit adjustment (Menon et al., 2021; Ren et al., 2020; Xu et al., 2021) for brevity (Equation 4). The $\pi_i$ is the statistical proportion of training set labels and we set $\tau = 1$ by default.

$$\hat{z}_i = z_i - \tau \cdot \log(\pi_i), \quad \pi_i = n_i / \sum_{j=1}^{C} n_j \tag{4}$$

Second, we depict the inference process in Figure 3 and outlined in Algorithm 1. Given the *top k* candidates, we retrieve the descriptor features $v^L$ from the cache and perform OVC with reweighted average similarities between the image and descriptor features.

$$z_i = \frac{1}{M} \cdot \sum_{m=1}^{M} W^{llm}_{\{i,m\}} \cdot \frac{v_i^V \cdot v_{\{i,m\}}^L}{||v_i^V||_2 \cdot ||v_{\{i,m\}}^L||_2} \tag{5}$$

The results show satisfactory interpretability (Figure 1c), as we can identify the model's decision-making basis by examining the similarity ranking of $M$ descriptor matching. We can also determine which fine-grained features the model misidentified that led to the final incorrect prediction.

---

**Algorithm 1** Inference pseudo code of HLC in a PyTorch-like style.

---

```
# Input: Visual_Encoder, p_vpt, W_cls(d x C), W_llm(C x M), D_llm(C x M x d), K, tau, pi
# Output: prediction y

for x in loader: # load images x from test set

    p_img = PatchEmbedding(x) # Project vanilla image to tokens

    p = torch.cat([p_vpt, p_img, p_cls], dim=1) # Add visual prompt tokens and CLS token

    v = Visual_Encoder(p) # Get image features from CLS token

    v_norm = v.norm(dim=-1, keepdim=True) # Get normalized feature for similarity

    z = W_cls @ v.T - tau * log(pi) # Calculate debiasing logits via Eq. 4

    idxs = torch.topk(z, K).indices # Get top K candidates classes

    # Calculate weighted similarity between v_norm and M descriptor features of each class
    score = [torch.mean(W_llm[i].softmax(dim=-1) * (v_norm @ D_llm[i].T)) for i in idxs]

    y = idxs[torch.argmax(torch.tensor(score))] # Get final decision from the K candidates
```

---

## 4 EXPERIMENT

### 4.1 DATASET

**Long-tailed visual datasets.** We conduct comprehensive experiments on 3 LTR benchmarks, namely Places-LT (Liu et al., 2019), ImageNet-LT (Liu et al., 2019), and iNaturalist 2018 (Horn et al., 2018). Places-LT is a long-tailed version of the large-scale scene recognition Places dataset (Zhou et al., 2017). It contains 62.5K images from 365 categories, with the instance number ranging from 5 to 4,980. ImageNet-LT is created by subsampling from ImageNet-2012 (Deng et al., 2009), consisting of 1,000 classes. The training set comprises 115.8K images, with the class image number ranging from 1,280 to 5. The validation and test sets are balanced, containing 20K and 50K images, respectively. iNaturalist 2018 (Horn et al., 2018) is a naturally long-tailed real-world dataset, comprising 8,142 fine-grained species and 437.5K images. We employ the official validation set for fair comparisons.

**Class-level text datasets.** We build the class-wise text corpus to fine-tune the classifier, visual prompt tokens, and descriptor cache weights for hierarchical classification. For the ***Text Corpus***, we utilize the corpus given by VL-LTR (Tian et al., 2022), which leverages class names as the query to retrieve entities from Wikipedia. After irrelevant section cleaning, we split the wiki sentences to construct the original text candidate set for each category. We incorporate handcraft-prompted (from CoOp (Zhou et al., 2022b)) and generated sentences (from CuPL (Pratt et al., 2022)) to achieve class balance. For ***Descriptor Cache***, we query the GPT-3.5-turbo for $M = 10$ fine-grained label descriptions. We request the LLMs to generate longer sentences instead of phrases to match the Text Corpus style. We utilize the CLIP text encoder to extract description features and cache them after normalisation to avoid loading the text encoder during inference.

### 4.2 COMPARISON WITH SOTA

We conduct comprehensive experiments on PlacesLT, ImageNet-LT and iNaturalist 2018 benchmarks. Our HLC outperforms state-of-the-art (full-finetuning methods) remarkably with minor parameters (visual prompt tokens and classifier) to tune. We provide detailed comparisons in terms of model size, tuning parameters, and inference required parameters. By default, we finetune the CLIP ViT-B-16 for 30 epochs with the initial learning rate $5e - 4$ and cosine decay schedule. We set the number of visual prompt tokens as 20 and the VPT layer depth $L_p = 12$. The sub-groups are split by instance number according to the SADE (Zhang et al., 2022).

**Comparisons on Places-LT.** Table 1 shows the experimental comparisons with previous SOTA on PlacesLT. The zero-shot CLIP does not require any fine-tuning but its performance is far from satisfactory. BALLAD (Ma et al., 2021) and VL-LTR (Tian et al., 2022) both fully fine-tuned CLIP using additional textual corpus and propose unique techniques to address the long-tail problem. However, our HLC outperforms them significantly without further complex LTR designs (only post-hoc logit adjustment Equation 4). Compared to VL-LTR, the HLC demonstrates significant advantages in

Table 1: Performance on the PlacesLT dataset. All methods are grouped by model type. VLMs refer to the dual encoder of vision (ViT-B/16) and text architecture. Our HLC achieves state-of-the-art results on all shots while requiring significantly fewer fine-tuning parameters.

| Method | Model | Tuning Params. | Model Params. | Many | Med. | Few | Acc. |
|---|---|---|---|---|---|---|---|
| OLTR (Liu et al., 2019) | | | | 44.7 | 37.0 | 25.3 | 35.9 |
| SADE (Zhang et al., 2022) | | | | 42.8 | 39.0 | 31.2 | 38.8 |
| MisLAS (Zhong et al., 2021) | ResNet152 | 60.34M | 60.34M | 39.6 | 43.3 | 36.1 | 40.4 |
| ALA (Zhao et al., 2022) | | | | 43.9 | 40.1 | 32.9 | 40.1 |
| PaCo (Cui et al., 2021) | | | | 36.1 | 47.9 | 35.3 | 41.2 |
| MAE (He et al., 2022) | | 111.66M | | 48.9 | 24.6 | 8.7 | 30.3 |
| DeiT III (Touvron et al., 2022) | | 86.66M | 86.66M | 51.6 | 31.0 | 9.4 | 34.2 |
| LiVT (Xu et al., 2023b) | ViT-B/16 | 111.66M | | 48.1 | 40.6 | 27.5 | 40.8 |
| VPT (Jia et al., 2022) | | 0.09M | 86.75M | 50.4 | 33.8 | 23.3 | 37.5 |
| LPT (Dong et al., 2023) | | 1.01M | 87.58M | 49.3 | **52.3** | 46.9 | 50.1 |
| RAC (Long et al., 2022) | | 86.57M | 236.19M | 48.7 | 48.3 | 41.8 | 47.2 |
| CLIP (Radford et al., 2021) | VLMs | 0M | | 35.0 | 37.3 | 44.2 | 37.9 |
| BALLAD (Ma et al., 2021) | | 149.62M | 149.62M | 49.3 | 50.2 | 48.4 | 49.5 |
| VL-LTR (Tian et al., 2022) | | 149.62M | | **54.2** | 48.5 | 42.0 | 50.1 |
| HLC (ours) | ViT-B/16 | 0.42M | 86.99M | 53.1 | 52.1 | **48.6** | **51.5** |

Table 2: Performance on the ImageNet-LT. Our HLC achieves state-of-the-art without backbone parameter tuning.

| Method | Many | Med. | Few | Acc. |
|---|---|---|---|---|
| with ResNet50 backbone | | | | |
| CE | 64.0 | 33.8 | 5.8 | 41.6 |
| c-RT | 61.8 | 46.2 | 27.3 | 49.6 |
| RIDE | 68.3 | 53.5 | 35.9 | 56.8 |
| PaCo | 68.0 | 56.4 | 37.2 | 58.2 |
| GCL | 63.0 | 52.7 | 37.1 | 54.5 |
| BCL | 67.6 | 54.6 | 36.6 | 57.2 |
| NCL | 67.3 | 55.4 | 39.0 | 57.7 |
| SADE | 66.5 | 57.0 | 43.5 | 58.8 |
| DLSA | 67.8 | 54.5 | 38.8 | 57.5 |
| with ViT-B / 16 backbone | | | | |
| DeiT III | 70.4 | 40.9 | 12.8 | 48.4 |
| LiVT | 73.6 | 56.4 | 41.0 | 60.9 |
| CLIP | 65.4 | 63.5 | 63.2 | 64.2 |
| VL-LTR | **84.5** | 74.6 | 59.3 | 77.2 |
| LPT | 76.6 | 73.3 | 67.6 | 73.7 |
| MARC+IVLM | 83.9 | 78.3 | 70.0 | 79.3 |
| HLC | 84.1 | **79.1** | **71.1** | **79.9** |

Table 3: Performance on the iNaturalist 2018. We report higher resolution results (@224 by default) for fair comparisons.

| Method | Many | Med. | Few | Acc. |
|---|---|---|---|---|
| with ResNet50 backbone | | | | |
| CE | 72.2 | 63.0 | 57.2 | 61.7 |
| OLTR | 59.0 | 64.1 | 64.9 | 63.9 |
| RIDE | 70.9 | 72.5 | 73.1 | 72.6 |
| TADE | 74.4 | 72.5 | 73.1 | 72.9 |
| PaCo | 75.0 | 75.5 | 74.7 | 75.2 |
| GCL | 67.5 | 71.3 | 71.5 | 71.0 |
| BCL | 66.7 | 71.0 | 70.7 | 70.4 |
| NCL | 72.0 | 74.9 | 73.8 | 74.2 |
| DOC | 72.8 | 71.7 | 70.0 | 71.0 |
| with ViT-B / 16 backbone | | | | |
| CLIP | 9.9 | 5.3 | 4.6 | 5.5 |
| LiVT | 78.9 | 76.5 | 74.8 | 76.1 |
| LPT | - | - | 79.3 | 76.1 |
| VL-LTR | 81.6 | 78.0 | 74.4 | 76.8 |
| VL-LTR@384 | - | - | - | 81.0 |
| HLC | 78.3 | 81.8 | 77.5 | 79.8 |
| HLC@336 | **79.1** | **81.8** | 80.6 | **81.1** |

terms of training epochs (30 *v.s.* 400) and tuning parameters (0.42M *v.s.* 149.62M). The LPT (Dong et al., 2023) has a similar number of tuning parameters to ours (1.01M *v.s.* 0.42M), while the longer input token sequence slows down its inference speed (Table 8). In contrast, our approach utilizes a few numbers of visual prompt tokens and conducts inference end to end.

**Comparisons on ImageNet-LT.** Table 2 presents the quantitative results on ImageNet-LT. Compared to previous works, multimodal methods (VL-LTR (Tian et al., 2022), IVLM (Wang et al., 2023)) show significant advantages for the excellent performance of CLIP on ImageNet. Both full fine-tuning and VPT methods ameliorate zero-shot CLIP, while our approach achieves state-of-the-art results. Note that our HLC is deployed without a text encoder, which ensures the model size is on par with multi-expert methods based on ResNet-50, such as NCL (Li et al., 2022a).

**Comparisons on iNaturalist 2018.** Table 3 presents the evaluation on the iNaturalist 2018 dataset. Due to the *domain gap between iNaturalist 2018 and CLIP training data*, direct zero-shot or linear probe yield unsatisfactory performance (Dong et al., 2023; Wang et al., 2023). Hence, we employ the full finetuning strategy (visual branch and classifier) to facilitate comparisons with other methods based on ImageNet21k. We fail to reproduce the LPT and VL-LTR@384 (we leave '-' in the table) because the training is overly complex for the 4× 2080Ti. Given the default resolution (@224), the HLC outperforms the previous SOTA remarkably. Note that the performance of HLC@336 is on par with VL-LTR@384 resolution.

Table 4: Ablation study on the PlacesLT based on CLIP (ViT/B-16). LP: linear probe. VPT: visual prompt tuning. LA: logit adjustment (Equation 4). Corpus: training with textual data. SFM: shared feature space mixup. Descriptor / Reweight: inference with (reweighted) feature descriptors.

| ID | Method | | | | | | Many | Med. | Few | Acc. |
|----|--------|--|--|--|--|--|------|------|-----|------|
| a) | Zero-shot CLIP | | | | | | 35.0 | 37.3 | 44.2 | 37.9 |
| b) | CLIP + Linear Probe | | | | | | 55.7 | 34.5 | 14.4 | 38.4 |
| c) | CLIP + Full Finetune | | | | | | 54.3 | 34.6 | 20.3 | 39.1 |
| | CLIP + Linear Probe | | | | | | | | | |
| ID | VPT | LA | Corpus | SFM | Descriptor | Reweight | Many | Med. | Few | Acc. |
| d) | ✓ | | | | | | **55.1** | 37.5 | 22.7 | 41.2 |
| e) | ✓ | ✓ | | | | | 50.2 | 47.2 | 40.7 | 47.1 |
| f) | ✓ | ✓ | ✓ | | | | 51.2 | 48.9 | 43.9 | 48.8 |
| g) | ✓ | ✓ | ✓ | ✓ | | | 53.1 | 50.2 | 46.4 | 50.5 |
| h) | ✓ | ✓ | ✓ | ✓ | ✓ | | 49.9 | 48.2 | 40.3 | 47.3 |
| i) | ✓ | ✓ | ✓ | ✓ | ✓ | ✓ | 53.1 | **52.1** | **48.6** | **51.5** |

Table 5: Performance comparisons with few-shot learning methods on the LTR and FSL datasets.

| Dataset | PlacesLT | | | | ImageNet-16 shot | |
|---------|----------|--|--|--|------------------|--|
| Method | Many | Med. | Few | Acc. | Acc. | Δ |
| CLIP | 35.0 | 37.3 | 44.2 | 37.9 | 72.43 | - |
| CoOp | 52.8 | 34.7 | 29.3 | 40.1 | 76.47 | 4.04 |
| CoCoOp | 50.0 | 33.4 | 33.0 | 39.3 | 75.98 | 3.55 |
| MaPLe | 53.8 | 36.2 | 29.5 | 41.2 | 76.66 | 4.23 |
| Ours | 52.7 | 41.1 | 36.8 | 44.5 | 77.06 | 4.63 |

Table 6: Ablation study of candidate number $K$ on the PlacesLT.

| Top K | Many | Med. | Few | Acc. |
|-------|------|------|-----|------|
| N/A | 52.7 | 41.1 | 36.8 | 44.5 |
| 1 | 53.1 | 50.2 | 46.4 | 50.5 |
| 3 | 53.1 | 50.6 | 46.9 | 50.8 |
| 5 | 53.3 | 51.4 | 48.3 | 51.5 |
| 7 | 52.6 | 49.7 | 46.2 | 50.1 |
| 9 | 52.5 | 49.1 | 45.1 | 49.5 |
| 11 | 52.3 | 47.8 | 43.2 | 48.5 |

## 4.3 ABLATION STUDY

In this paper, we construct text corpora and feature description cache to assist VLMs. We propose shared feature space mixup (SFM) and weighted feature descriptors to align image features with language. Besides, we employ VPT (Jia et al., 2022) and LA (Menon et al., 2021) on top of the baseline to enhance fine-tuning effectiveness. Hence, we conduct experiments on PlacesLT to verify the efficacy of all proposals and the results are presented in Table 4. Combined with Table 1, we observe that the zero-shot CLIP has surpassed the previous baseline performance (OLTR (Liu et al., 2019) 35.9% vs. CLIP 37.9%), but fails to catch up with SOTA (*e.g.*, LPT 50.1%). Considering the types **a-c**, the linear probe and full fine-tuning exhibit similar performance (38.4% vs. 39.1%), albeit with a significant difference in optimization parameters. Consequently, we adopt the CLIP + linear probe in the following experiments. The effectiveness of VPT is corroborated by type **d**, which indicates that VPT inspires better adaptation of the downstream distribution without compromising the CLIP feature extraction capability (Dong et al., 2023). Type **e** demonstrates that LA can effectively calibrate the model's prior biases (see sub-group performance), which serves as a concise post-processing approach. Type **f** and **g** demonstrate that the multimodal data consistently improves overall performance, particularly on the few-shot subgroup. Mixed with high-quality textual features, the tail image features have better anchor (centre) representation, and our SFM further facilitates the convergence of tail image features. Directly utilizing the average similarity of descriptors results in performance degradation (type **g** and **h**), which is incurred by the erroneous or non-informative descriptions that do not contribute to the visual recognition (Figure 6). We optimize each description weight to mitigate the aforementioned adverse effects (type **i**).

## 4.4 FURTHER DISCUSSION

**Comparisons with FSL methods.** Our HLC is inspired by few-shot learning (FSL). Therefore, we reproduce the FSL methods, *e.g.*, CoOp (Zhou et al., 2022b), CoCoOp (Zhou et al., 2022a) and MaPLe (khattak et al., 2023) based on the MaPLe code repository. We conduct sufficient experiments on both LTR and few-shot benchmarks and present the results in Table 5. *We retain only VPT and reweighted descriptors* of the HLC to ensure comparable complexity with FSL methods. The descriptors' weights are trained on the LTR benchmarks. From Table 5, the FSL methods have consistently improved the zero-shot CLIP performance on both datasets. Our HLC significantly outperforms FSL methods on the LTR dataset (+3.3% compared to MaPLe), thereby providing com-

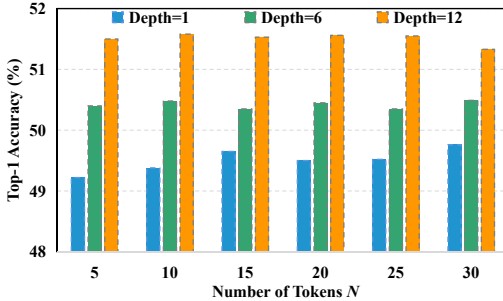 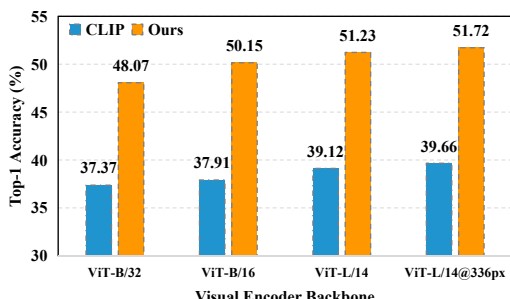

Figure 4: Ablation study of VPT tokens ($N$) and layer depth ($L_p$) on the PlacesLT. We adopt ViT-B/16 as the backbone for all settings.

Figure 5: Ablation study of the visual encoder backbone on the PlacesLT. @336px means the input image size is $336 \times 336$.

Table 7: Zero-shot OVC performance on the PlacesLT. †: GPT-3. ‡: GPT-3.5-turbo. We further reweight the descriptors given by LLMs.

| Method | CLIP | VCD† | VCD‡ | Ours‡ |
|---|---|---|---|---|
| Acc. | 37.91 | 40.34 | 40.51 | 43.12 |
| $\Delta$ | - | +2.43 | +2.6 | +5.21 |

Table 8: Inference performance on the iNaturalist 2018 with batch size 64. We outperform SOTA in computation and reasoning time.

| Method | VL-LTR | LPT | HLC |
|---|---|---|---|
| FLOPs (T) | 1.301 | 2.406 | 1.189 |
| Inf. time (ms) | 354.17 | 513.62 | 269.32 |

pelling evidence for the crucial role of fine-grained descriptors and corresponding weights. Our approach also demonstrates its effectiveness on the ImageNet 16-shot base classes. Our text branch has no learnable tokens and relies on the basic prompt `"a photo of {CLS}"`. Nevertheless, our proposal achieves superior performance compared to MaPLe (76.66% *v.s.* 77.06%).

**Number of Candidates $K$.** Table 6 shows the effect of the candidate number $K$ given by the LTR classifier. N/A means that we directly calculate the reweighted average similarity for all classes (no LTR classifier). $K = 1$ means the LTR classifier performance. Note that larger $K$ can not always lead to better performance. More candidate categories introduce more noise information. The $K$ mainly affects med and few groups. Hence, we set $K = 5$ by default for our experiments.

**Number of Prompt Tokens $N$ and Depth $L_p$.** We conduct experiments on the PlacesLT to investigate the impact of visual prompt tokens $N$ and layer depth $L_p$. As shown in Figure 4, the impact of $L_p$ on the performance is significant, while the impact of token number $N$ is minor. Note that the maximum $N$ is 30, as longer token sequences will remarkably slow down the inference speed. Therefore, we set $N = 20$ and $L_p = 12$ by default for other experiments.

**Effect of Visual Encoder.** Figure 5 demonstrates the effect of different visual encoders. The backbone shows minimal effect on both zero-shot CLIP and proposed HLC. In contrast, proper text features will serve as effective anchors to guide model classification. This conclusion aligns with prior works such as CoOp and VCD.

**Comparisons with VCD.** VCD (Menon & Vondrick, 2023) is the first to utilize the LLMs (GPT-3) to generate fine-grained descriptions for assisting VLMs in open vocabulary classification. Our HLC is different in 2 aspects: 1) we employ more advanced LLMs (GPT-3.5-turbo) to generate more apt descriptors; 2) we reweight each description to filter out the irrelevant ones. The comparisons are shown in Table 7. Both descriptions provided by GPT-3 and GPT-3.5-turbo enhance the performance of zero-shot CLIP. However, the reweighting operation significantly improves the OVC performance, demonstrating its success in mitigating the influence of irrelevant descriptions on visual recognition.

**Inference Analysis.** Table 8 shows the model FLOPs and inference time for an epoch. We evaluate the validation dataset of iNaturalist 2018 with batch size 64 on 2080Ti. Based on ViT-B-16, our HLC is much smaller on both FLOPs and average inference time than the previous SOTA.

**Failure Case Analysis.** Why is it necessary to reweight descriptors? We observe that some descriptors are not beneficial for visual recognition and thus result in lower average V-L similarity. These cases encompass descriptions that do not contribute to visual recognition (Fig. 6a: description of the sound.) and that are inconsistent with the images in the benchmark datasets (Fig. 6b: no signs are

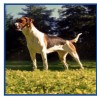
**Label**: English Foxhound

**Descriptor**: English Foxhounds have a distinctive, melodious bay that can be heard for miles.

(a) Correct but unuseful.

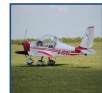
**Label**: Airfield

**Descriptor**: Airfield may have different markings and signs that help pilots navigate.

(b) Correct but mismatch.

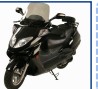
**Label**: Vespa

**Descriptor**: Vespa wasps have a body shape with a narrow waist and a bulbous abdomen.

(c) Word ambiguity.

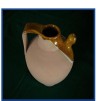
**Label**: Pitcher

**Descriptor**: Tea or water: a pitcher is typically made of tea or water.

(d) Error description.

Figure 6: Examples of 4 type failure descriptors given by large language models. Our reweighting operation effectively mitigates the negative influence of types (a) and (b).

present in the category *Airfield*.). By reweighting, we can decrease the similarity of such descriptors and emphasize the ones that are truly useful for visual recognition. There are also failure cases due to label word ambiguity (Fig. 6c: mistakenly interpreting the commercial brand "Vespa" as a type of wasp) and factual inaccuracies resulting from hallucinations (Fig. 6d). However, we believe these issues can be alleviated along with LLMs' rapid development.

## 5 RELATED WORK

**Long-tailed Visual Recognition**. The most straightforward approach is to address the issue through rebalancing techniques, which encompass resampling via balanced or inverse sampler (Cao et al., 2019; Kang et al., 2020; Zhang et al., 2021; Li et al., 2022b; Dong et al., 2023) and reweighting via loss weight (Cui et al., 2019; Tang et al., 2020; Tan et al., 2020) or margin (Menon et al., 2021; Xu et al., 2023b; Li et al., 2022b) methods. The LTR data augmentation enhances the tail samples via feature mixing (Chou et al., 2020; Chu et al., 2020) or generation (Li et al., 2021; Park et al., 2022). The Mixture of Experts (MoE) method proposes to learn different parts of LTR data (Wang et al., 2021; Li et al., 2022a; Jin et al., 2023; Xu et al., 2023a).

**Visual Language Models** have been proposed to facilitate downstream tasks by introducing extra language data (Radford et al., 2021; Alayrac et al., 2022; Jia et al., 2021). CoOp (Zhou et al., 2022b) learns soft text prompts to improve the zero-shot CLIP performance. CoCoOp (Zhou et al., 2022a) formulates the text prompts with instance-level conditions. VPT (Jia et al., 2022) introduces visual prompts to effectively fine-tune the visual branch to fit the downstream data distribution. MaPLe (khattak et al., 2023) jointly optimizes the visual and text prompts and employs mapping to establish a correspondence between the two types of prompts. The CuPL (Pratt et al., 2022), VCD (Menon & Vondrick, 2023) and CHiLS (Novack et al., 2023) have further leveraged large language models to generate fine-grained descriptions of class labels to enhance the text branch of VLMs. These FSL methods hold implications for learning from tail classes.

**Learning LTR data with Visual Language Models**. The impressive zero-shot capabilities exhibited by VLMs have inspired a series of excellent works on long-tail recognition. The VL-LTR (Tian et al., 2022) conducts full fine-tuning of CLIP (Radford et al., 2021), followed by a language-guided recognition head to adapt the long-tail data. Similarly, BALLAD (Ma et al., 2021) leverages a linear adapter to mitigate the impact of long-tail bias. LMPT (Xia et al., 2023) introduces an embedding loss with class-aware soft margin and re-weighting to learn class-specific contexts. IVLM (Wang et al., 2023) incorporates a lightweight decoder to accommodate previous work on LTR and provides a comprehensive assessment of their performance based on VLMs.

## 6 CONCLUSION

In this paper, we propose a hierarchical long-tailed classification (HLC) framework to address the long-tailed recognition problem. We employ visual prompt tuning and propose the shard space mixup to train an effective coarse classifier. Then, we utilize large language models to generate fine-grained descriptions for each class and train corresponding weights to filter out irrelevant ones. Given *top k* candidate classes from the coarse classifier, we perform fine-grained open vocabulary classification based on descriptions. Our approach achieves state-of-the-art performance with minimal parameters to tune and enhances the interpretability of the prediction results.

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
