# OpenReview forum: "Hierarchical Long-tailed Classification with Visual Language Models"
_ICLR.cc/2024/Conference — ICLR 2024 Conference Withdrawn Submission_

### Official Review · Reviewer_2kUk · 2023-10-26

**Soundness:** 2 fair
**Presentation:** 2 fair
**Contribution:** 2 fair
**Rating:** 5
**Confidence:** 4

**Summary:**

This paper studies the problem of adapting existing vision language models (VLMs) to long-tailed data. They especially propose a hierarchical framework that prioritizes candidate categories before conducting fine-grained classification using detailed textual descriptions. Specifically, they first incorporate visual prompt tokens and leverage shared feature space mixup for multimodal feature interactions. Based on candidates given by the coarse classifier, they query large language models to generate corresponding fine-grained descriptions to refine the final predictions. They also consider a reweighting mechanism to filter out invalid descriptions generated by language models. The experiment results on commonly used datasets (PlacesLT, ImageNet-LT, and iNaturalist 2018) show the effectiveness of the proposed method with only a few tuned parameters.

**Strengths:**

1. The experiment results are comprehensive. With only a few tuned parameters, the proposed method achieves very good performance across different long-tailed datasets.
2. The presentation of the experiment parts is clear.
3. The structure of the paper is clear and easy to follow.

**Weaknesses:**

1. The motivation of the paper is not strong enough.  As claimed in the intro: "while previous methods can achieve satisfactory top-5 classification accuracy, the true challenge lies in fine-grained predictions from the candidates".

The fine-grained predictions are actually the top-1 accuracy, which is the most commonly used criterion in long-tailed recognition.

2. Then, the authors raise the problem that "Unfortunately, while OVC excels at handling fine-grained recognition with label prompts (Yao et al., 2021), it is incapable of catching up with the state-of-the-art approaches (Table 1) in the challenging LTR, because the OVC fails to 1) take advantage of existing annotation information; 2) overcome the long-tail bias problem; and 3) fully encapsulate the visual information with existing language description."

There are two problems. First, previous methods already studied the problem of adapting existing OVC models to long-tailed datasets and showed that with proper strategies, the final obtained model can achieve good performance across long-tailed datasets [1,2]. Second, the authors attribute the reasons why existing OVC models do not perform well in long-tailed datasets to three points. For example, the first reason is that "fail to take advantage of existing annotation information", however, no further explanation of how the model fails to take advantage of annotation information and no related evidence is provided to support the analysis. They directly go to illustrate how they solve the problem.

3. In the proposed method, it is more likely the combination of different existing strategies. The motivation for some operations is not very clear. For example, when introducing the usage of deep Visual Prompt Tuning (VPT), the motivation is "Considering the similarity between the few-shot learning and tail learning of LTR", there are many other strategies to solve few-shot learning, why only VPT is adopted and why do not adopt other strategies?

4. In section 3.2, the authors talk about the "To perform fine-grained classification, we further construct descriptor sets that characterize M detailed features of the category. ", this part matches with the motivation in the introduction. Then, what is the point of section 3.1?

5. I am also confused about how "construct descriptor sets that characterize M detailed features of the category." will lead to fine-grained classification. The motivation is not clear.

6. In the introduction, the author keeps repeating how they perform their method in paragraph 4 and paragraph 5. However, the necessary motivation and explanation are limited.

Basically, the paper is easy to follow, but the writing can be further improved.  Other suggestions are in the "Questions" part.

[1] Changyao Tian, Wenhai Wang, Xizhou Zhu, Jifeng Dai, and Yu Qiao. VL-LTR: learning class-wise visual-linguistic representation for long-tailed visual recognition.

[2] Bowen Dong, Pan Zhou, Shuicheng Yan, and Wangmeng Zuo. LPT: Long-tailed prompt tuning for image classification. In ICLR, 2023

**Questions:**

Some suggestions on the writing:

1. From Sec 3.1 to 3.2, it would be much better if several sentences could be added to explain why fine-grained classification is necessary.
2. In the related works, it is necessary to discuss the difference between the proposed method and existing methods.

---

### Official Review · Reviewer_yLRq · 2023-10-27

**Soundness:** 1 poor
**Presentation:** 1 poor
**Contribution:** 2 fair
**Rating:** 3
**Confidence:** 4

**Summary:**

This paper introduces pre-trained vision-language models to enhance long-tailed classification. It proposes a hierarchical framework (HLC) to first predict the candidate classes, and then query detailed descriptions and refine the final fine-grained predictions. For the candidate class predictions, HLC utilizes visual prompt tuning to parameter-efficiently fine-tune the CLIP model, as well as a shared feature space mixup to enhance the modality interaction. For the fine-grained prediction, HLC introduces learnable weights for each description, considering their different relationships to the images. Experimental results demonstrate the effectiveness of the proposed method.

**Strengths:**

1. This paper studies long-tailed classification with vision-language models, where the multi-modal information could be utilized to improve the tail class performance.
2. This paper proposes some new methods including HLC, SFM, and reweighted descriptions. Experiments demonstrate the effectiveness of these modules.
3. The proposed method achieves a state-of-the-art performance in long-tailed classification.

**Weaknesses:**

1. I am concerned with the reproducibility of this work:

   a) The authors have not released the source code. To reproduce the results, some important implementation details are missed, such as the optimizer (SGD or Adam or other?), the batch size, the weight decay (if has), and the momentum...

   b) The descriptions for classes are not released. Since the answers from GPT contain randomness even if you give a fixed question, it is necessary to open-source the descriptions for each class. However, the authors only provide limited descriptions in the supplementary materials.

2. Some statements lack evidence:

   a) In the introduction, the authors claim that "naively enumerating possible fine-grained classifiers and training each candidate yield exponential computational overheads". This lacks evidence.

   b) The authors claim that the classifier performs better on textual modality than on images. This is not experimentally verified.

3. The proposed method is too complex. It contains multiple components, including a) visual prompts, b) feature space mixup, c) logit debiasing, and d) text corpus, descriptors, and description weights. However, the motivation for combining these modules is relatively weak.

   I suggest the authors add more explanations as to why the hierarchical framework is necessary for long-tailed classification, and why you need to incorporate the above components, instead of merely proposing such an intricate framework.

   Moreover, the comparison of top-1 and top-5 accuracy in Fig 1(a) is not convincing, since for almost all methods, top-5 accuracy is naturally higher than top-1 accuracy. I suggest the authors conduct more observations. For instance, you can illustrate that fine-grained predictions are too weak (maybe) when compared with coarse-grained predictions.

4. The experimental settings are inconsistent. On Places-LT and ImageNet-LT, HLC utilizes visual prompt tuning. On iNaturalist 2018, however, HLC fully finetunes the visual branch and classifier. Moreover, it is compared with methods based on ImageNet21k. Models from CLIP or ImageNet21k might have different abilities.

5. The writing needs improving.

   a) In the introduction, it is written that "instead of training numerous candidate classifiers, we alternatively employ OVC for further fine-grained classification without parameter tuning (Figure 1b)." However, the statement is not related to Figure 1b.

   b) The dataset name Places-LT is not unified. Sometimes it is written as "PlacesLT".

   c) In Figure 3, "loigts" should be "logits".

**Questions:**

Apart from the "weakness", I have the following questions.

1. Considering the modality gap, the textual and visual features might be distanced. Does this have an effect on the Shared Feature space Mixup (SFM)?
2. How to ensure that the learned weights for each descriptor are optimal? Since some classes have limited data, it is hard to learn the corresponding description weights.
3. What is the difference between the effects of text corpus and descriptor cache? Why not directly generate descriptions for all classes and apply the fine-grained classification?
4. How to run HLC@336 with ViT-B/16 backbone? As far as I know, CLIP does not release ViT-B/16 with 336×336 resolution (it only releases ViT-L/14 with 336×336 resolution). Since the authors do not open-source the code, it is hard to reproduce the results.

---

### Official Review · Reviewer_LavU · 2023-10-28

**Soundness:** 3 good
**Presentation:** 3 good
**Contribution:** 3 good
**Rating:** 6
**Confidence:** 4

**Summary:**

This paper proposes HLC, a hierarchical long-tailed framework with Vision Language Models (VLMs). During training, it adopts classifier fine-tuning with visual-prompt tuning and mixs up concatenated multi-modal features generated by the CLIP encoders. During inference, it uses fine-grained descriptions generated by large language models to further reweight the predicted top-k logits from the trained coarse classifier. As a result, HLC achieves SoTA performance on common LTR benchmarks with fewer tuned parameters.

**Strengths:**

- The paper is easy to follow.

- The experiments results are strong with detailed ablations.

- Using LLMs for generating textual desciptions are interesting.

**Weaknesses:**

- Only ViTs are considered as backbone, while the results of using ResNet as many previous methods did are missing.

- The overall pipeline is a little complicated, and some of its components already exist (e.g. VPT, LA, Corpus), which may limit the contribution of this work.

**Questions:**

1. What is the influence of training batch size considering the batch-level SFM?

2. What is the performance of using descriptions from different source (e.g. textual prompts, wikipedia corpus)?

---

### Official Review · Reviewer_CVWh · 2023-11-01

**Soundness:** 3 good
**Presentation:** 2 fair
**Contribution:** 2 fair
**Rating:** 5
**Confidence:** 4

**Summary:**

This paper focuses on the long-tailed recognition task by effectively using knowledge from vision-language models (CLIP). The method consists of two parts. They first using a cross-modal prompt tuning strategy for vision encoder. Then in inference stage, they leverage large language models and learnable weights to build descriptors for further classification. The effectiveness of the proposed method is evaluated on Places-LT, ImageNet-LT and iNaturalist 2018. The results on benchmarks seem promising.

**Strengths:**

1. The motivation, hierarchical classification from coarse to fine-grained, is clear and promising. And the method proposed is also closely developed around the motivation.

2. Compared with previous related work, this work requires fewer learnable parameters. Therefore, from the perspective of model learning and updating, the training cost would be relatively lower.

3. It performs well on the benchmark dataset.

**Weaknesses:**

1. The necessary discussion on multi-modal mixup is lacking. The effectiveness of multi-modal mixup in downstream tasks has been confirmed by some works [1,2] to be superior to single-modal mixup, and here it seems that existing tools are just used to enhance the interaction between modalities. But what are the benefits of doing this? Apart from the help in performance, what deeper connection does it have with imbalanced classification? For example, would such a multi-modal mixup make the learned visual features more balanced, thus providing significant help for tail classes?

[1] Geodesic Multi-Modal Mixup for Robust Fine-Tuning
[2] MixGen: A New Multi-Modal Data Augmentation

2. The introduction to the symbols used is missing. For example, dimension? The lack of this information causes the reader to spend a lot of time calculating and guessing what the meaning and information behind the symbols are during reading.

3. The description of ERM needs to be confirmed again. Equation 1 seems to only describe how to obtain the prediction but does not involve how to reduce the difference between the prediction and the ground truth. This is not entirely consistent with what ERM describes. Therefore, another issue arises. It seems that no loss function was mentioned. It should be cross-entropy. Use CE loss to make the mixed concat features and mixed labels close, thereby updating $\mathbf{W}^{cls}$?


4. What is $\mathcal{L}$ in equation 3? A transform layer? Why use the shallow-VPT of the first p layers? What are the advantages of doing this? Regarding the choice of p, does the setting of the long-tail problem provide some principal selection basis? For example, p=1, p=3 is enough for tail classes to learn a good representation? Beyond the VPT, there are also additional parameter-efficient tuning methods of CLIP. I would like to see how the proposed method performs when combined with other tuning methods, such as adapter, lora, adapterformer etc.

5. Although the method of using a large language model to generate some fine-grained class-level descriptions and use them to update the model was not first proposed in this paper, such an attempt is still meaningful and interesting. Moreover, prompting the model with descriptions generated by the LLM, obtaining fine-grained features, and using them to achieve better classification is closely related to motivation and meaning. I hope to see more discussions on related work.

6. The logit adjustment and re-weight strategies used in Section 3.3 are from existing work. I didn’t ask that every part must be original, but there is a lack of necessary discussion and details here. For example, the initialization of $(\mathbf{W}^{llm})$? Is there a better estimation form possible?

7. The necessary introduction to the algorithm during the training phase, loss function, and other key information are missing. If there is not enough space in the main text, it should at least be placed in the appendix. But I didn’t see it, and it seems that there is no such introduction in the supplementary materials.

8. The experimental part lacks many implementation details.  Considering that no source code is provided, this raises concerns about the reproducibility of the work.

9. In Table 4, type h gives a performance of 47.3%, which is significantly lower than LPT, BALLAD, and VL-LTR. This seems to suggest that the proposed method is not sufficiently effective. This could potentially be due to the descriptor containing too many erroneous descriptions, and the learning of the re-weighting strategy has improved the model’s generalization. On the other hand, this also increases the burden of the method. It requires not only additional data but also strategies to correct the negative impact brought by the additional data.

10. Looking forward to more detailed experiments and discussions.

**Questions:**

Please refer to the weakness part.